# Triple Isozyme Lactic Acid Dehydrogenase Inhibition in Fully Viable MDA-MB-231 Cells Induces Cytostatic Effects That Are Not Reversed by Exogenous Lactic Acid

**DOI:** 10.3390/biom11121751

**Published:** 2021-11-24

**Authors:** Elizabeth Mazzio, Nzinga Mack, Ramesh B. Badisa, Karam F. A. Soliman

**Affiliations:** 1Institute of Public Health, College of Pharmacy & Pharmaceutical Sciences, Florida A&M University, Tallahassee, FL 32307, USA; elizabeth.mazzio@famu.edu (E.M.); Nzinga.mack@jhu.edu (N.M.); ramesh.badisa@famu.edu (R.B.B.); 2 Institute of Computational Medicine, Johns Hopkins Whiting School of Engineering, Baltimore, MD 21218, USA

**Keywords:** LDH, glycolysis, inhibitor, cancer cells

## Abstract

A number of aggressive human malignant tumors are characterized by an intensified glycolytic rate, over-expression of lactic acid dehydrogenase A (LDHA), and subsequent lactate accumulation, all of which contribute toward an acidic peri-cellular immunosuppressive tumor microenvironment (TME). While recent focus has been directed at how to inhibit LDHA, it is now becoming clear that multiple isozymes of LDH must be simultaneously inhibited in order to fully suppress lactic acid and halt glycolysis. In this work we explore the biochemical and genomic consequences of an applied triple LDH isozyme inhibitor (A, B, and C) (GNE-140) in MDA-MB-231 triple-negative breast cancer cells (TNBC) cells. The findings confirm that GNE-140 does in fact, fully block the production of lactic acid, which also results in a block of glucose utilization and severe impedance of the glycolytic pathway. Without a fully functional glycolytic pathway, breast cancer cells continue to thrive, sustain viability, produce ample energy, and maintain mitochondrial potential (ΔΨ_M_). The only observable negative consequence of GNE-140 in this work, was the attenuation of cell division, evident in both 2D and 3D cultures and occurring in fully viable cells. Of important note, the cytostatic effects were not reversed by the addition of exogenous (+) lactic acid. While the effects of GNE-140 on the whole transcriptome were mild (12 up-regulated differential expressed genes (DEGs); 77 down-regulated DEGs) out of the 48,226 evaluated, the down-regulated DEGS collectively centered around a loss of genes related to mitosis, cell cycle, GO/G1–G1/S transition, and DNA replication. These data were also observed with digital florescence cytometry and flow cytometry, both corroborating a G0/G1 phase blockage. In conclusion, the findings in this work suggest there is an unknown element linking LDH enzyme activity to cell cycle progression, and this factor is completely independent of lactic acid. The data also establish that complete inhibition of LDH in cancer cells is not a detriment to cell viability or basic production of energy.

## 1. Introduction

Triple-negative breast cancer (TNBC) is an aggressive cancer, most common in African American women, being associated with fewer treatment options, advanced stage diagnosis, high recurrence, and low overall survival rates [1,2]. While a wide variety of solid tumors display elevated aerobic glycolysis, concomitant to over-expressed lactic acid dehydrogenase (LDH) and production of lactic acid, these factors are accentuated in TNBC compared with hormone receptor-positive breast cancers [3,4]. Worse, primary taxane-based chemotherapies used to treat TNBC contribute toward a glycolytic-rich chemo-resistant acidic phenotype in which LDHA expression is amplified [5,6]. Although the pursuit of LDHA inhibitors has been underway, there appears to be a gap in the literature exploring the consequences of inhibiting the enzyme itself vs. blocking production of the end product, where many studies never measured for changes in lactic acid. In our facility, we found that the inhibition of LDHA failed to block lactic acid production or alter glycolysis in diverse cancer cell line models; including (1) cells with LDHA silenced genes using lentiviral shRNA and [7] (2) applying confirmed human recombinant LDHA inhibitors derived from a high-throughput screening, which had no effect on lactic acid produced, glucose consumed, or ATP when tested in vitro [8,9]. These findings led to the supposition that (1) either there is another biochemical route to the production of lactic acid or (2) more than one LDH isozyme (isoform) is responsible for lactic acid production, of which the latter hypothesis was confirmed. Confirmation came from observing the biochemistry taking place in an established dual knock out of both LDHA and LDHB, which fully blocked lactic acid, attenuated glucose use, and severely impeded glycolysis [10,11]. Therefore, a full block of lactic acid production by inhibiting several forms of LDH is absolutely pivotal in setting forth a proper experimental design that seeks to define the consequences of targeting LDH in cancer, in particular, as it relates to the full shut down of glycolysis. It was not until the recent availability of multiplex LDH isoform inhibitors (LDHA, LDHB, LDHC, and LDHD) or combined silencing models—where lactic acid in a cancer model was blocked for the first time—enabling study of how cancer cells survive without lactic acid acidity, which independently is a known integral role in metastasis and immune evasion [12,13].

In 2016, a research team cataloged GNE-140 from a high throughput screening of two million compounds as having ability to inhibit LDHA, LDHB, and LDHC with IC_50_s in the nanomolar range, which could entirely block lactic acid production in cancer cells [14]. In this work, we investigate the effects of GNE-140 on biochemical parameters and whole transcriptomic response in a TNBC cell model.

## 2. Materials and Methods

### 2.1. Materials

Dulbecco’s modified Eagle’s medium (DMEM), fetal bovine serum (FBS), penicillin/streptomycin, general reagents and supplies were purchased from Millipore-Sigma (St. Louis, MO, USA) and VWR International (Radnor, PA, USA). All microarray equipment, reagents, and materials were purchased from Affymetrix/Thermo Fisher Scientific (Waltham, MA, USA).

### 2.2. Cell Culture

MDA-MB-231 cells were purchased from American Type Cell Culture (ATCC) (Manassas, VA, USA). Cells were cultured in 75 cm^2^ flasks containing DMEM, supplemented with 10% FBS and 10,000 U/mL penicillin G sodium/10,000 μg/mL streptomycin sulfate. Cells were grown at 37 °C with 5% CO_2_ atmosphere and sub-cultured every three to five days.

### 2.3. Cell Viability

Cells were plated in 96-well plates at 0.5 × 10^5^ cells per well (24 h toxicity assay) and 0.04 × 10^5^ cells per well (7 day proliferation study). After experiment endpoints, resazurin (Alamar Blue) (Millipore-Sigma, St. Louis, MO, USA) indicator dye was used to measure viable cell count. Briefly, a working solution of resazurin (0.5 mg/mL) was prepared in sterile PBS; filter sterilized through a 0.2 micron filter; added to the samples (15% (*v*/*v*) equivalent); and returned to the incubator for 2–6 h. Reduction of the dye by viable cells reduced the oxidized resazurin, yielding a bright red fluorescent intermediate resorufin, quantified using a Synergy multi-mode reader (Model HTX, Bio-Tek (Agilent), Winooski, VT, USA) using 530 nm (excitation)/590 nm (emission) filters. Spheroids were grown in the ultra-low attachment (ULA) 96-well plates, which enable the self-assembly of spheroids grown to maturation for 7 days achieving uniformity in size and shape. After maturation, GNE-140 was added to the wells, and spheroids were grown for another 14 days followed by image analysis using phase contrast and fluorescence digital microscope photography using a Biotek Cytation 5 imaging Station and Gen 5.0 software (Winooski, VT, USA).

### 2.4. Imaging—Mitochondria, Morphology, and Nuclear Staining

Monolayer imaging: Morphological cell imaging of monolayers was acquired in fixed permeabilized cells stained with Phalloidin-iFluor 488 Reagent [ab176753] Abcam (Cambridge, MA, USA) and a propidium iodide (PI) nuclear counterstain (Millipore-Sigma, St. Louis, MO, USA). Briefly, cells were fixed in 4% paraformaldehyde for 20 min, washed, and then permeabilized with 0.2% Triton X-100, (Millipore-Sigma, St. Louis, MO, USA) in water, and stained for 30 min. DAPI staining was obtained in live cells reflecting reduced cell count in time extended proliferation studies. Live mitochondrial imaging was acquired using MitoTracker™ Red; Thermo Fisher Scientific (Waltham, MA, USA). Stock solutions containing fluorescent probes and nuclear stains were prepared by following the manufacturer guidelines or dissolving by weight/volume; 5 mg/1 mL ethanol, with subsequent dilutions made in HBSS prior to be added to cells; working final dye concentrations: 5 µg/mL PI, DAPI, or 6.6 µM phalloidin. Spheroid imaging: Live cell imaging of spheroids was carried out with phase-contrast imaging, merged with fluorescent images of fluorescein diacetate (FDA) staining using a Biotek-Agilent Cytation 5 (Winooski, VT, USA). Briefly, a stock solution of FDA was prepared by dissolving 5 mg/mL in acetone (250×), and dilutions were freshly prepared by diluting in PBS, added to the samples, and incubated for 40 min to an hour at 37 °C at 5% CO_2_/Atm.

### 2.5. Glucose and Lactic Acid

Determination of lactic acid was carried out using a colorimetric enzymatic assay, according to the manufacturer’s instructions (Trinity Biotech, Jamestown, NY, USA). Lactic acid quantification was carried out in phenol red-free/low serum media to minimize cross-interference with the reagent. Briefly, the reagent was added to the samples, incubated for approximately 8 min, and quantified at 490 nm using a Bio-Tek/Agilent Synergy multi-mode reader Model HTX (Winooski, VT, USA). Glucose was quantified as previously using an enzymatic assay containing equal volume of glucose oxidase (20 U/mL) and a chromogenic solution containing 1 mM vanillic acid, 500 μM of 4-aminoantipyrine, and four purpurogallin units per milliliter of horseradish peroxidase type II. Glucose was quantified at 550 nm on a Bio-Tek Synergy HTX multi-mode reader.

### 2.6. Cytometry

Flow Cytometry: Cells were cultured in 75 cm_2_ flasks at 0.5 × 10^6^ cells/mL in the absence or presence of 50 μM GNE-140. After 24 h, cells were trypsinized and centrifuged at 1268× *g* for 4 min. Then the cell-pellets were resuspended in 100 μL PBS, fixed in 5 mL of 100% ethanol (pre-cooled at −20 °C), added dropwise to each tube while vortexing, and incubated at 4 °C for 30 min. In order to remove the fixative, the cells were centrifuged (1268× *g*) for 4 min, pellets re-suspended in 100 μL PBS and vortexed gently. Finally, 0.75 mL staining solution was added per sample in the dark and incubated at room temperature for 1 h with occasional stirring. The distribution of DNA in all cell-cycle phases was assayed within 1 h by using the Becton Dickerson FACS Calibur flow cytometer (BD Biosciences, San Jose, CA, USA). In each sample, a total of 20,000 individual events were analyzed. CellQuest Pro software (BD Biosciences, San Jose, CA, USA) was used for the acquisition and analysis of the data, and the percentage of cells in each phase was determined by using ModFit LT 3.0 software (Verity Software House, Topsham, ME, USA).

Digital Fluorescence Cytometry: Digital Cytometry was carried out in accordance to Bioteks technical notes for the Cytation 5 imaging station (entitled: Cell Synchrony Determination Using Microscopic Imaging Use of Nuclear Staining to Assess Cell Cycle Stage by Nuclear DNA Content: Application Note/Paul Held Ph. D). Briefly, cells were synchronized for 24 h in low serum media, then placed in 7% FBS supplemented growth media, and treated with various concentrations of GNE-140 in a 96-well plate. After 3 days, cells were analyzed using DAPI and stained for 45 min at 37 °C −5% CO_2_/atmosphere. Analytical settings and processes included the following: acquisition at 10× objective lens, 12 photo montage, followed by montage stitching, image preprocessing (background flattening, rolling ball diameter 743 micron), and analysis with initial object threshold set to 5000 and minimum object size 5–25 µM. Cut off values for G1, S, and G2/M phase were set empirically based on object data obtained from untreated control wells, and set under sub-population histogram parameters for mean intensity values.

### 2.7. Microarray WT Human Datasets

After the experimental time point, cells were scraped, washed three times in ice-cold HBSS, spun down, the supernatant removed, and the remaining pellet was rapidly frozen and stored at −80 °C. Total RNA was isolated and purified using the Trizol/chloroform method. The RNA quality was assessed, and concentrations were equalized to 82 ng/μL in nuclease-free water. According to the GeneChipTM WT PLUS Reagent Manual for Whole Transcript (WT) expression arrays, whole transcriptome analysis was conducted. Briefly, RNA was reverse transcribed to first-strand or second-strand cDNA, followed by cRNA amplification and purification. After the 2nd cycle of ss-cDNA Synthesis and hydrolysis of RNA, ss-cDNA was assessed for yield, fragmented, labeled, and hybridized onto the arrays, before being subjected to fluidics and imaging using the Gene Atlas (Affymetrix, ThermoFisher Scientific (Waltham, MA, USA). The array data quality control and initial processing from CEL to CHP files were conducted using an expression console, prior to data evaluation using the Affymetrix transcriptome analysis console, the STRINGdb protein–protein interaction networks functional enrichment analyses and the Database for Annotation, Visualization, and Integrated Discovery (DAVID, Bioinformatics Resources 6.8. located at https://david.ncifcrf.gov, accessed on 18 November 2021.

### 2.8. Data Analysis

Statistical analysis was performed for basic studies using Graph Pad Prism (version 3.0; Graph Pad Software Inc., San Diego, CA, USA). The significance of the difference between the groups was assessed using either a Student’s *t*-test or a one-way ANOVA and followed by Tukey’s post-hoc analysis.

## 3. Results

The discovery of GNE-140 originated from a high-throughput screening of two million small molecules, which resulted in the elucidation of an effective pan-LDH inhibitor with the capacity to inhibit LDHA/B/C in the low *n*M range—GNE-140 chemical structure (Figure 1A) which by crystal structure of the drug, comes into complex with the human LDH pocket (Figure 1B) [14].

In this work, the effects of GNE-140 were evaluated over a concentration dose range in MDA-MB-231 cells with data being collected for lactic acid produced and glucose consumed. The findings in this study clearly demonstrate that GNE-140 can effectively block lactic acid production, inverse to hampered utilization of glucose, suggesting an attenuated rate of glycolysis (Figure 2A). This manipulation had no adverse effects on cell viability or mitochondrial membrane potential, but there was a loss of cell division, as observed by a 7 day end-point proliferation study (Figure 2B), also shown in Figure 3 images, which was associated with enlargement of the actin cytoskeleton in static cells that were unable to divide (Figure 4). 

To determine if the attenuation of lactic acid concentration plays a contributing role in the cytostatic properties of GNE-140, the 7 day proliferation study was repeated at the IC_50_ for GNE (10 μM), adding increasing concentrations of (+) lactic acid (Figure 5). The data in Figure 5 suggest that the cytostatic properties of this drug appear to be related directly to LDH enzyme activity (and loss of glycolysis), completely independently of the loss of the end product—lactic acid.

GNE-140 was also effective in preventing the growth of 3D spheroids over a 2 week growth period (Figure 6), where the viability as ascertained with FDA staining shows a cytostatic tumor in the presence 120 μM of GNE-140, results consistent with the lack of toxicity observed in the monolayer (Figure 2A), or morphological integrity of the monolayer (Figure 6 (right-hand panel)). Of note, is that the higher fluorescence in smaller tumors treated with GNE-140 is likely due to loss of cytoskeletal dynamics, as shown in Figure 4, with greater penetration of the dye relative to tightly compact larger spheroids, which would be the case for the untreated controls.

The concentration chosen to evaluate GNE-140 was 50 μM, where glycolysis and cell proliferation are both near fully halted, without a loss in cell viability. Figure 7 provides an overall summary of the Affymetrix transcriptomic array data. In brief, 48,226 genes were tested, arrays were run 2 times (Set 1: Controls/*n* = 2, GNE-140/*n* = 3) (Set 2: Controls/*n* = 3, GNE-140/*n* = 2), and the sets were cross-compared. The data selection for criteria included the following: repeatability in both sets for changed direction and significance *p*-value < 0.05 with a fold change (FC) less than, or greater than, 2. The cutoff for FDR *p*-value was <0.1. The effects of GNE-140 were very mild compared with what is normally observed with drugs when examining the whole transcriptome. In this set, we see only 12 up-regulated DEGS and 77 down-regulated DEGs out of the 48226 non-protein-coding microRNA, long-intergenic RNAs, and protein-coding mRNAs tested. The data is presented in Table 1 by gene symbol, gene description, fold change, and *p*-Value.

To ascribe the basic functional systems affected by GNE-140 at 50 µM on the whole transcriptome, data was evaluated using String database analysis, as presented for down-regulated transcripts (Figure 8) and up-regulated transcripts (Figure 9).

The overall database analytics presented by String_db (Figure 8) align with KEGG pathway analysis (cell cycle) determined using the Database for Annotation, Visualization, and Integrated Discovery (DAVID) Figure 10. The genomics work was corroborated by flow and digital cytometric analysis (Figure 11 and Figure 12, respectively).

## 4. Discussion

A wide variety of solid tumors display abnormal metabolic patterns obviated by rapid utilization of glucose, overexpression of LDH proteins, and excess production of lactic acid compared to normal tissue. Our study investigated the consequences of inhibiting multiple isoforms of LDH with a single reagent (GNE-140), on MDA-MB-231 TNBC cells, showing total suppression of lactic acid production, severely stunted glucose utilization, and an impediment to what is termed the anaerobic glycolytic pathway.

The data from this study provide insight into a number of pre-existing questions about the relevance of LDH in cancer. First, there was an unclear role about the ramifications of inhibiting LDH, where many inhibitors either fail to reduce lactic acid or have a toxicity profile in close proximity to inhibition of LDH, such as gossypol. Given that dead cells do not produce lactic acid, despite LDH inhibition or not, we could not ascertain any concrete association between LDH inhibition with cell death or LDH inhibitors and reduction in lactic acid. Moreover, gossypol, as the most successful LDH inhibitor candidate, reached phase I/II trials in humans, showing no therapeutic effects with no partial or complete responses among 20 women receiving gossypol for metastatic breast cancer refractory to doxorubicin and paclitaxel [15].

Similar to gossypol, an LDHA inhibitor, we found that gene silencing of LDHA in TNBC cells had no effect on lactic acid, glucose utilization, or energy. From this, we surmised that multiple silencing or inhibition of multiple LDH isoforms might be needed to block lactic acid, which turned out to be the case. LDH/AB dual silencing can fully block lactic acid in fully viable cells, having no apparent negative effect on cell viability or energy systems. At this point, we can clearly see that ramification of LDH inhibition entirely not only is not cytotoxic to cancer cells but in some instances can re-invigorate mitochondrial respiration, while at the same time, it corresponds to slower rates of cell division. Having these new research tools and multiplex LDH-inhibitor drugs, such as GSK2837808A and GNE-140, which fully block lactic acid, suggests that in the absence or presence of lactic acid, slower rates of cell proliferation are continually realized [14,16,17].

That said, the only adverse anticancer effect of LDH inhibition appears to be in attenuation of cell proliferation, independent of (+) lactic acid production. The anti-mitotic effect of various LDH inhibitors (of independent isoforms) has been observed in diverse models [18], some using natural LDH inhibitors such as saffron [19,20], diallyl trisulfide (DATS) [21], galloflavin derivative’s [22] PGG, novel synthetic bifunctional nanoparticles, and drugs [23,24,25]. LDH inhibition causing anti-mitotic effects is consistently reported in a variety of cancer models [24,26], including 2D or 3D cultures and xenograft transplants in nude mice [27,28,29,30,31] and, in many instances, can augment the efficacy of chemo- or radiation-resistant cancers [32,33,34] and impede EMT, migration, or clonogenicity [35]. The findings in this work also corroborate the original work of Boudreau et al. investigating the effects of GNE-140 on human PaCa-2 cancer cells; with a block in lactic acid production, concomitant to sparse glucose utilization and anti-mitotic effects at similar concentrations [14].

A number of overarching questions emerge as to the relationship or mechanism of action between LDH enzyme function and cell cycle control, which needs further investigation. It is plausible that blocking LDH activity severely attenuates glucose use and may slow the generation of glycolytic carbon intermediates that feed alternative pathways to promote macromolecule synthesis needed for cellular division. Moreover, if inhibiting LDH can, in some cases, strengthen the respiratory capacity of non-dividing cancer cells, this could foster stem cell phenotype and lead to propagation of a metabolic rewiring, which could later lead to invasive metastasis [11,36], creating counter indication of LDH inhibitor drugs. [11] The latter question is currently being addressed by research investigating combination use of LDH inhibitors with mitochondrial OXPHOS or the Krebs cycle-inhibiting drugs [37,38] either together or adjunct to traditional chemotherapy drugs such as paclitaxel [39,40].

While many questions about the Warburg effect remain, in theory, one may rationalize that the use of multiplex LDH inhibitors would have therapeutic attributes as there would be a reduction in lactic acidosis surrounding tumors, which alone drives metastasis, angiogenesis, and immune escape. LDH inhibitors that could block lactic acid would also acquiesce the propelling role of acidity in driving loss of immune surveillance, a lower infiltration of CD3+ and CD4+ T cells, infiltration of tumor-associated macrophages (TAMs), lower response to anti-PD1 inhibitors [41,42], and acid-mediated macrophage polarization, corresponding to advanced metastatic migration and invasion [43].

## 5. Conclusions

In conclusion, the data from this study confirm that inhibiting multiple isoforms of LDH in a cancer cell will effectively shut down rapid glucose use through glycolysis and block lactic acid production, which is not cytotoxic. Also, current results and the previous literature show some controlling elements of the robust nature of the LDH enzyme being in control over the cell cycle and being independent from the presence or absence of lactic acid. Future studies will be required to define precisely how LDH activity controls the cell cycle, albeit through severely stunted glucose utilization or a direct control on cell cycle regulatory components.

## Figures and Tables

**Figure 1 biomolecules-11-01751-f001:**
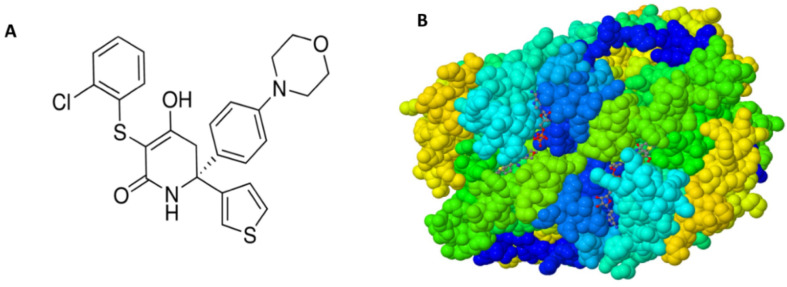
(**A**) Chemical Structure of GNE 140, a selective human lactate dehydrogenase (LDH) inhibitor (LDH-A, LDH-B, and LDH-C) and (**B**) Lactate dehydrogenase A in complex with a trisubstituted piperidine-2,4-dione-inhibitor GNE-140 (RCSB PBD). Source: https://www.rcsb.org/structure/4ZVV; Deposited: 18 May 2015, Released: 18 May 2016. Deposition Author(s): Li, Y., Chen, Z., Eigenbrot, C. doi:10.2210/pdb4ZVV/pdb: Image was created using *Mol RCSB protein databank from publication [14] and Viewer: modern web app for 3D visualization and analysis of large biomolecular structures. Nucleic Acids Research. doi:10.1093/nar/gkab314.

**Figure 2 biomolecules-11-01751-f002:**
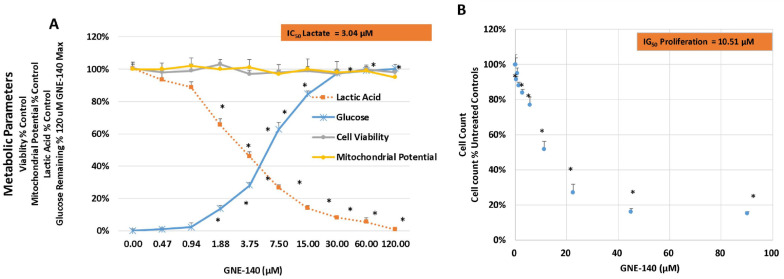
Biochemical effects of GNE-140 in MDA-MB-231 cells. Cell viability, lactic acid, and mitochondrial potential were tested at 24 h, while glucose consumption was tested when the differential between the media blank and the cell controls reached ~90% of total glucose consumed (approximately 72 h). (**A**) Cell proliferation was conducted after 7 days of incubation. (**B**) The data are represented as % control and expressed as the mean ± S.E.M, *n* = 4. * *p* < 0.05. The IC_50_ for lactic acid inhibition was 3.04 µM, where the inhibitory growth (IG)_50_ for cell proliferation was 10.51 µM.

**Figure 3 biomolecules-11-01751-f003:**
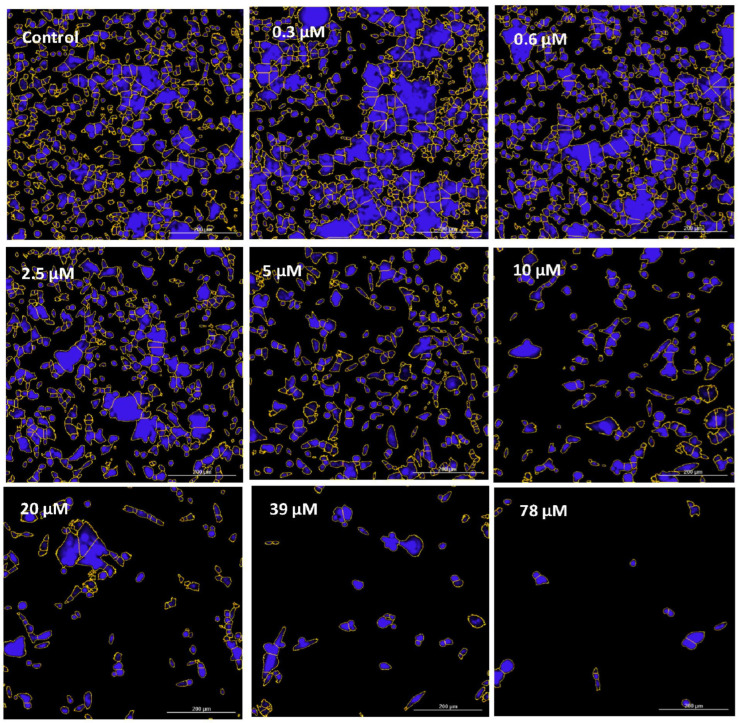
Cytostatic effects of GNE-140 in MDA-MB-231 cells after a 7 day proliferation period. Images represent nuclear cell count in viable cells using DAPI staining, with cell count analytical masking (yellow outline) established with the Biotek Cytation 5 Analytical Imaging System (Winooski, VT, USA). The images corroborate the cytostatic properties of the drug, as shown in Figure 2B, using Alamar Blue detection agent for cell counting.

**Figure 4 biomolecules-11-01751-f004:**
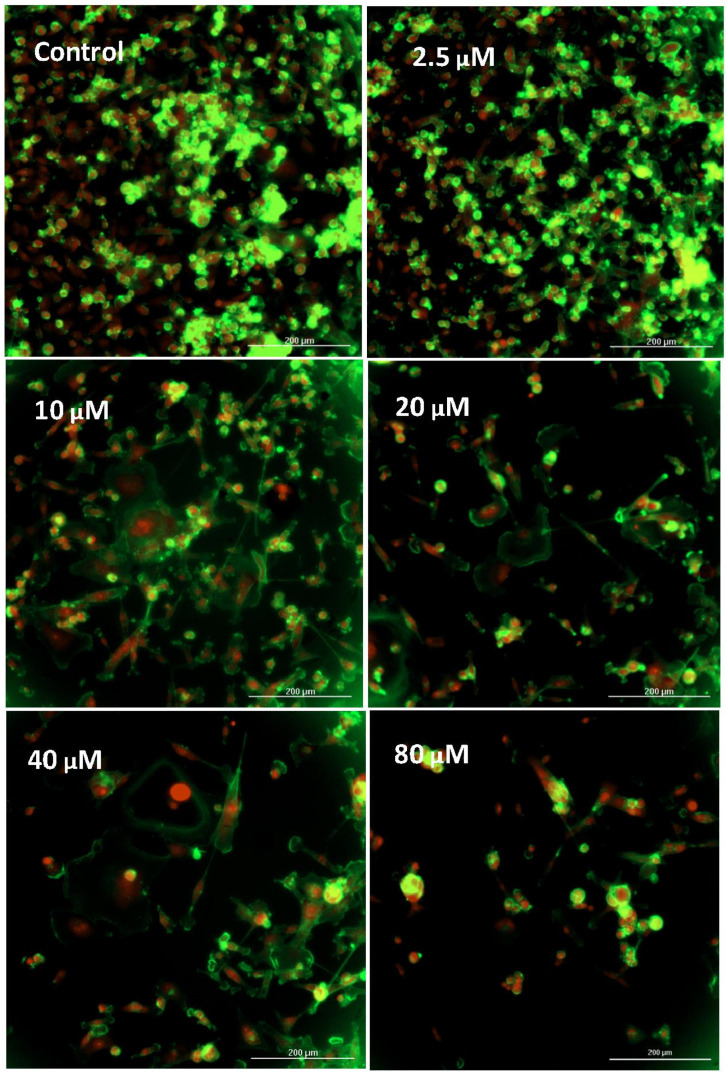
Cytoskeletal actin structural changes corresponding to the cytostatic effects of GNE-140 in MDA-MB-231 cells after a 7 day proliferation period. Images represent fixed, permeabilized cells which were stained with phalloidin 488 and a nuclear PI counter stain.

**Figure 5 biomolecules-11-01751-f005:**
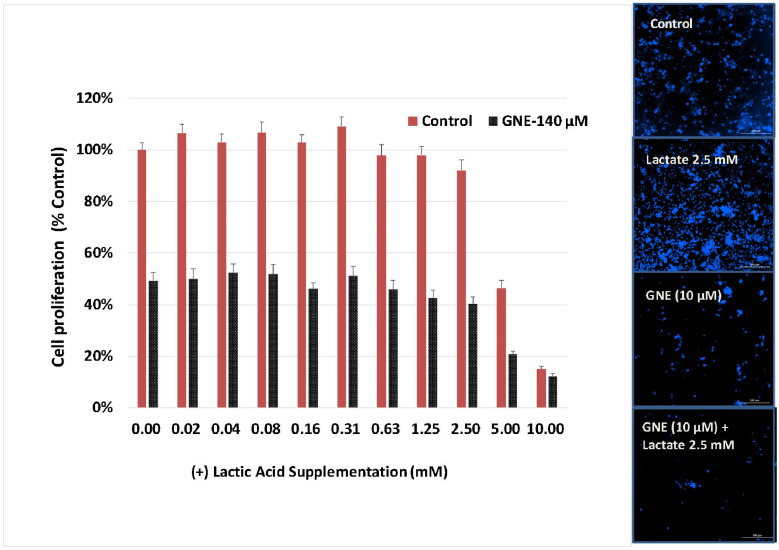
Non-reversible cytostatic effects of GNE-140 at 7 days in the presence or absence of exogenous supplemented (+) lactic acid in the media. The data represent cell proliferation and are expressed as % untreated controls, with GNE-140 (10 µM) held constant in combination with increasing concentration of (+) lactic acid. Viable cell count was measured using Alamar Blue (left panel) and images taken using DAPI nuclear staining (right panel). The data are expressed as the mean ± S.E.M, *n* = 4.

**Figure 6 biomolecules-11-01751-f006:**
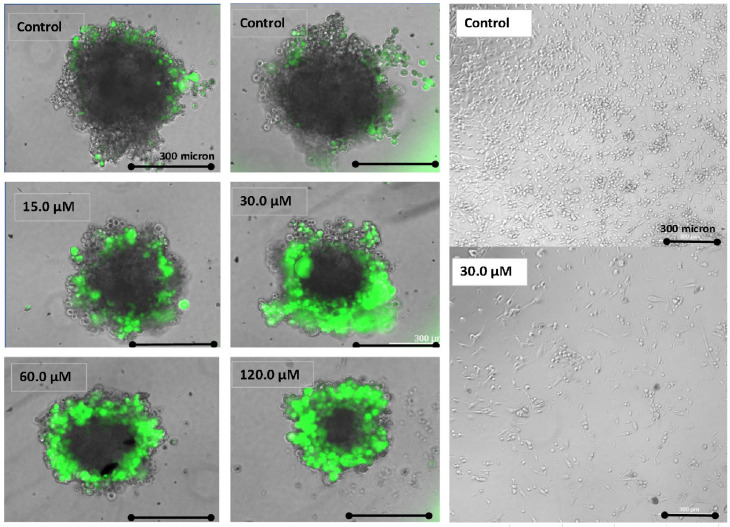
Cytostatic effects of GNE-140 in MDA-MB-231 3D cells spheroids, after 2 weeks of incubation (left-hand panel). Live spheroid imaging was obtained using fluorescein diacetate (FDA) merged with phase contrast images. The right-hand panel confirms 2D monolayer morphology in live static cells when treated with GNE-140 (30 µM) after 7 day proliferation. These findings confirm that the loss of LDH does not impact energy, but rather the mitotic division of cells, selectively.

**Figure 7 biomolecules-11-01751-f007:**
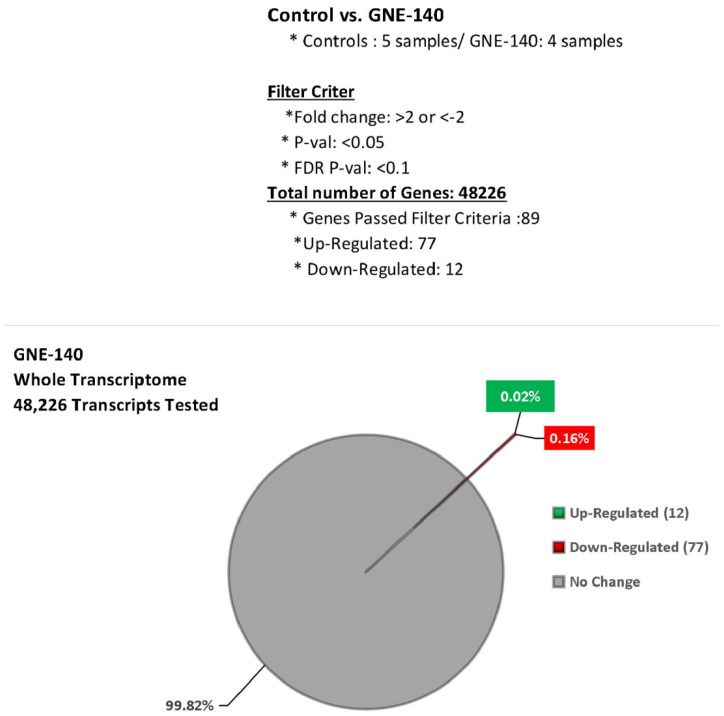
Transcriptome summary data for the effects of GNE-140 vs. controls in MDA-MB-231 cells.

**Figure 8 biomolecules-11-01751-f008:**
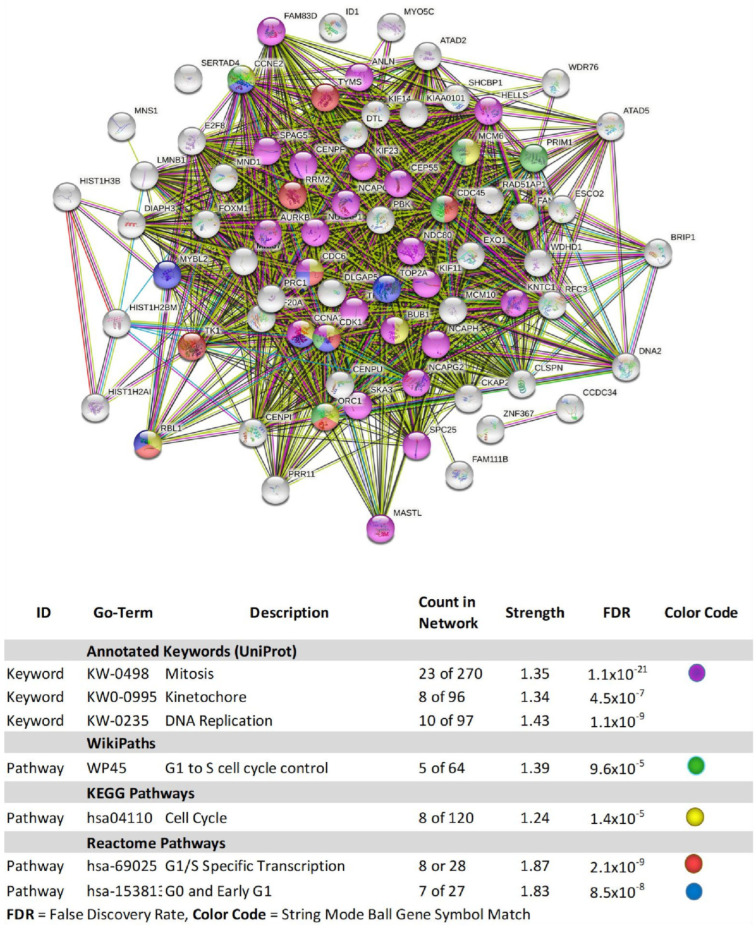
String_db classification of down-regulated DEGs by GNE-140 in MDA-MB-231 cells. The diagram represents gene symbols, which are color-coded by category in the database analysis table. Table data contains database used, ID within each specific database, description of pathway, count in network, strength of association, FDR *p*-value, and color code for genes involved in that network.

**Figure 9 biomolecules-11-01751-f009:**
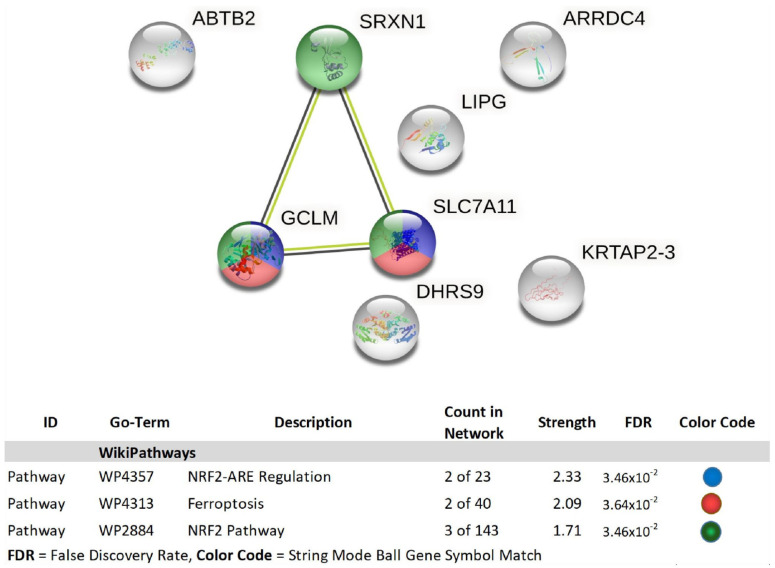
Classification of up-regulated DEGs by GNE-140 in MDA-MB-231 cells, determined using Protein–Protein Interactive String Database analysis. The diagram data are presented as gene symbol and are color-coded by category in the database analysis table. String analytical tables provide the database used, the ID within that database of the recognized related change, the description of the pathway, count in network, strength of association, FDR *p*-value, and color code for genes involved in that network.

**Figure 10 biomolecules-11-01751-f010:**
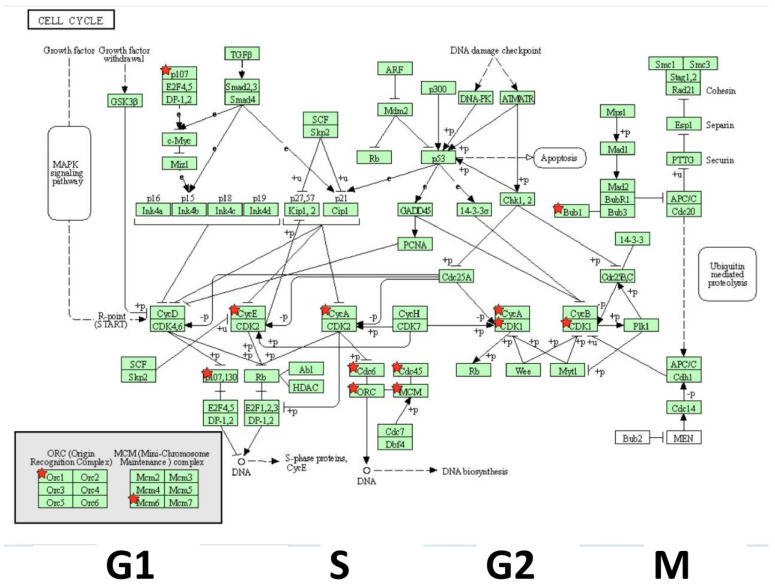
DEG-down-regulated transcripts by GNE-140 vs. controls in MDA-MB-231 cells by pathway analysis show loss of function in cell cycle related genes. Down-regulated DEGS are denoted (red stars) within the KEGG cell cycle pathway.

**Figure 11 biomolecules-11-01751-f011:**
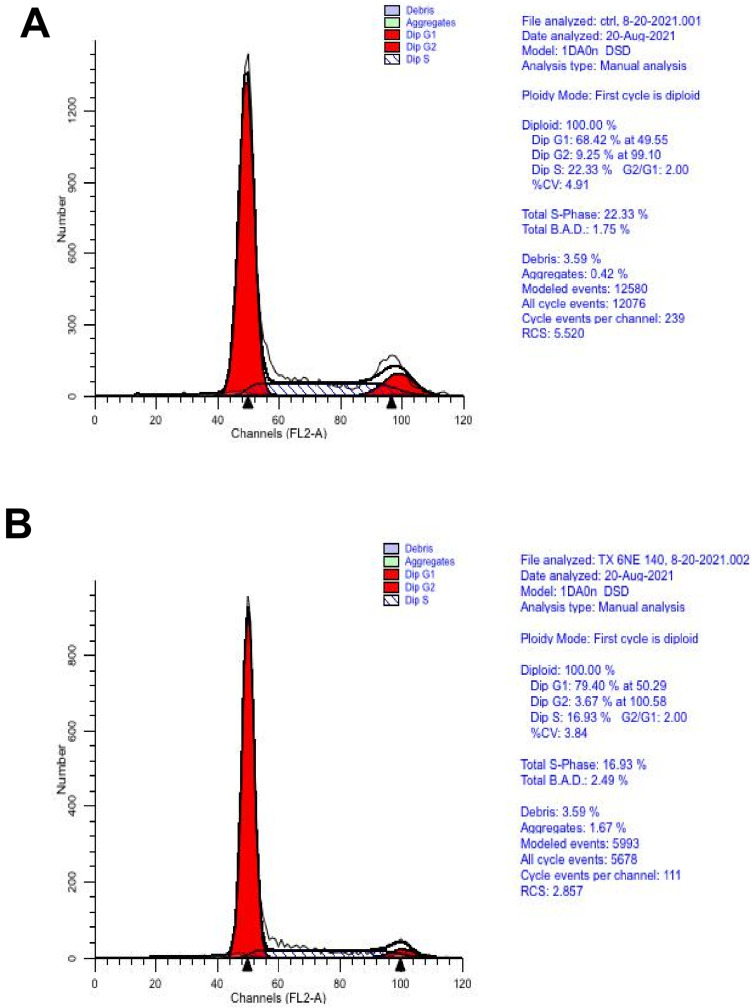
The effects of GNE-140 (50 µM) on cell cycle progression in MDA-MB-231 cells as determined by flow cytometry (**A**) Control vs. GNE-140 treatment (**B**). Abbreviations: 1D0n DSD is the name of the program run for the sample under auto-analysis of the software; total B.A.D—background aggregates and debris (<20% is good); RCS—reduced chi-square; DIP—diploid; %CV—coefficient of variance, <8 is acceptable.

**Figure 12 biomolecules-11-01751-f012:**
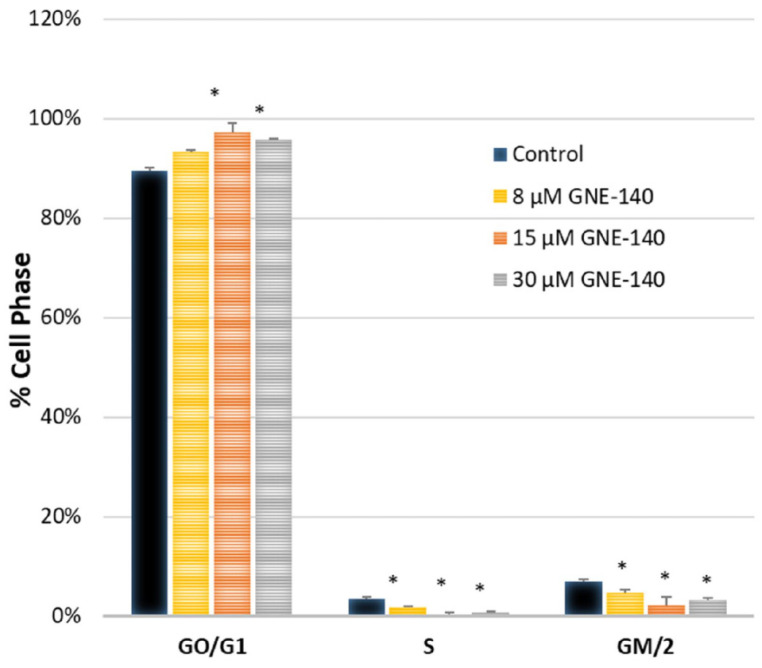
Digital cell cycle analysis in GNE-140-treated MDA-MB-231 cells was performed using fluorescence cytometry. The data represents the average ± the mean of cells at various stages, as a % of total cell cycle phases, *n* = 3. The significance of differences between the groups was determined by a one-way ANOVA with Tukey’s post-hoc test. * *p* < 0.05.

**Table 1 biomolecules-11-01751-t001:** Transcriptome shift in GNE-140 treated cells vs. controls.

	Gene	Description	FC	*p*-Value	FDR *p*-Value
1	*SNORA38B*	small nucleolar RNA, H/ACA box 38B	−6.59	2.58 × 10^−5^	0.0259
2	*BRIP1*	BRCA1 interacting protein C-terminal helicase 1	−4.65	1.31 × 10^−5^	0.0257
3	*HIST1H3B*	histone cluster 1, H3b	−4.40	6.50 × 10^−6^	0.0257
4	*PBK*	PDZ binding kinase	−4.19	1.30 × 10^−5^	0.0257
5	*NDC80*	NDC80 kinetochore complex component	−3.94	3.70 × 10^−6^	0.0257
6	*NCAPG*	non-SMC condensin I complex subunit G	−3.91	9.67 × 10^−6^	0.0257
7	*FANCI*	Fanconi anemia complementation group I	−3.78	4.27 × 10^−5^	0.0355
8	*HIST1H2AI*	histone cluster 1, H2ai	−3.62	3.74 × 10^−5^	0.0328
9	*EXO1*	exonuclease 1	−3.51	1.09 × 10^−5^	0.0257
10	*ANLN*	anillin actin binding protein	−3.49	8.98 × 10^−6^	0.0257
11	*TYMS*	thymidylate synthetase	−3.49	8.01 × 10^−6^	0.0257
12	*SHCBP1*	SHC SH2-domain binding protein 1	−3.46	3.54 ×10^−5^	0.0322
13	*FAM111B*	family with sequence similarity 111, member B	−3.40	2.00 ×10^−4^	0.0676
14	*HIST1H2BM*	histone cluster 1, H2bm	−3.39	6.37 ×10^−5^	0.0436
15	*CDK1*	cyclin-dependent kinase 1	−3.35	1.47 ×10^−5^	0.0257
16	*DIAPH3*	diaphanous-related formin 3	−3.28	1.24 × 10^−5^	0.0257
17	*MYBL2*	v-myb avian myeloblastosis viral oncogene homolog-like 2	−3.28	1.62 × 10^−5^	0.0257
18	*KIF11*	kinesin family member 11	−3.27	2.55 × 10^−5^	0.0259
19	*HELLS*	helicase, lymphoid-specific	−3.16	2.00 × 10^−4^	0.0665
20	*RRM2*	ribonucleotide reductase M2	−3.11	1.37 × 10^−5^	0.0257
21	*CENPU*	centromere protein U	−3.09	1.78 × 10^−5^	0.0257
22	*CCNA2*	cyclin A2	−3.06	2.40 × 10^−5^	0.0257
23	*TOP2A*	topoisomerase (DNA) II alpha	−3.01	1.50 × 10^−5^	0.0257
24	*SKA3*	spindle and kinetochore associated complex subunit 3	−2.99	1.83 × 10^−5^	0.0257
25	*CCNE2*	cyclin E2	−2.97	4.83 × 10^−5^	0.0388
26	*ESCO2*	establishment of sister chromatid cohesion N-acetyltransferase 2	−2.94	6.92 × 10^−5^	0.0445
27	*RAD51AP1*	RAD51 associated protein 1	−2.91	2.00 × 10^−4^	0.0727
28	*CLSPN*	claspin	−2.87	2.45 × 10^−5^	0.0257
29	*MKI67*	marker of proliferation Ki−67	−2.85	7.56 × 10^−5^	0.0457
30	*TK1*	thymidine kinase 1, soluble	−2.85	1.76 × 10^−5^	0.0257
31	*MCM10*	minichromosome maintenance 10 replication initiation factors	−2.83	1.97 × 10^−5^	0.0257
32	*NUSAP1*	nucleolar and spindle associated protein 1	−2.80	1.78 × 10^−5^	0.0257
33	*SPAG5*	sperm associated antigen 5	−2.78	7.33 × 10^−5^	0.0457
34	*SERTAD4*	SERTA domain containing 4	−2.75	2.00 × 10^−4^	0.0665
35	*MNS1*	meiosis-specific nuclear structural 1	−2.74	7.71 × 10^−5^	0.0457
36	*DNA2*	DNA replication helicase/nuclease 2	−2.73	1.00 × 10^−4^	0.0525
37	*CDC6*	cell division cycle 6	−2.71	4.93 × 10^−5^	0.0389
38	*DLGAP5*	discs, large (Drosophila) homolog-associated protein 5	−2.70	6.80 × 10^−5^	0.0443
39	*KIF14*	kinesin family member 14	−2.70	2.68 × 10^−5^	0.0264
40	*CEP55*	centrosomal protein 55kDa	−2.69	3.39 × 10^−5^	0.0315
41	*ATAD5*	ATPase family, AAA domain containing 5	−2.68	6.59 × 10^−5^	0.0439
42	*CDC45*	cell division cycle 45	−2.67	9.80 × 10^−5^	0.0518
43	*ATAD2*	ATPase family, AAA domain containing 2	−2.64	4.48 × 10^−5^	0.0367
44	*NCAPH*	non-SMC condensin I complex subunit H	−2.61	8.88 × 10^−5^	0.049
45	*CCDC34*	coiled-coil domain containing 34	−2.61	3.88 × 10^−5^	0.0328
46	*MND1*	meiotic nuclear divisions 1	−2.55	3.00 × 10^−4^	0.0911
47	*DTL*	denticleless E3 ubiquitin protein ligase homolog (Drosophila)	−2.54	5.08 × 10^−5^	0.0391
48	*NCAPG2*	non-SMC condensin II complex subunit G2	−2.48	3.85 × 10^−5^	0.0328
49	*SPC25*	SPC25, NDC80 kinetochore complex component	−2.48	6.39 × 10^−5^	0.0436
50	*KNTC1*	kinetochore associated 1	−2.43	5.36 × 10^−5^	0.0391
51	*WDR76*	WD repeat domain 76	−2.42	7.45 × 10^−5^	0.0457
52	*E2F8*	E2F transcription factor 8	−2.37	7.77 × 10^−5^	0.0457
53	*WDHD1*	WD repeat and HMG-box DNA binding protein 1	−2.36	6.64 × 10^−5^	0.0439
54	*CKAP2L*	cytoskeleton associated protein 2-like	−2.35	5.35 × 10^−5^	0.0391
55	*MASTL*	microtubule associated serine/threonine kinase-like	−2.34	6.22 × 10^−5^	0.0436
56	*CENPI*	centromere protein I	−2.32	1.00 × 10^−4^	0.0601
57	*PRIM1*	primase, DNA, polypeptide 1 (49kDa)	−2.31	7.12 × 10^−5^	0.0452
58	*FAM83D*	family with sequence similarity 83, member D	−2.29	1.00 × 10^−4^	0.0542
59	*HIST2H3A; HIST2H3C*	histone cluster 2, H3a; histone cluster 2, H3c	−2.28	1.00 × 10^−4^	0.0575
60	*KIF23*	kinesin family member 23	−2.27	1.00 × 10^−4^	0.0575
61	*MCM6*	minichromosome maintenance complex component 6	−2.27	1.00 × 10^−4^	0.0575
62	*ZNF367*	zinc finger protein 367	−2.25	8.12 × 10^−5^	0.0466
63	*LMNB1*	lamin B1	−2.23	1.00 × 10^−4^	0.0601
64	*BUB1*	BUB1 mitotic checkpoint serine/threonine kinase	−2.22	8.93 × 10^−5^	0.049
65	*ID1*	inhibitor of DNA binding 1, dominant negative helix-loop-helix protein	−2.21	1.00 × 10^−4^	0.0534
66	*CENPF*	centromere protein F	−2.21	9.11 × 10^−5^	0.0493
67	*KIAA0101*	KIAA0101	−2.21	2.00 × 10^−4^	0.0653
68	*AURKB*	aurora kinase B	−2.20	2.00 × 10^−4^	0.0651
69	*TPX2*	TPX2, microtubule-associated	−2.18	1.00 × 10^−4^	0.0525
70	*ORC1*	origin recognition complex subunit 1	−2.18	2.00 × 10^−4^	0.0653
71	*RBL1*	retinoblastoma-like 1	−2.17	2.00 × 10^−4^	0.0789
72	*FOXM1*	forkhead box M1	−2.17	1.00 × 10^−4^	0.0525
73	*KIF20A*	kinesin family member 20A	−2.16	3.00 × 10^−4^	0.0885
74	*MYO5C*	myosin VC	−2.10	3.00 × 10^−4^	0.0842
75	*PRC1*	protein regulator of cytokinesis 1	−2.09	2.00 × 10^−4^	0.0772
76	*PRR11*	proline rich 11	−2.05	2.00 × 10^−4^	0.0651
77	*RFC3*	replication factor C subunit 3	−2.04	2.00 × 10^−4^	0.0667
1	*GCLM*	glutamate-cysteine ligase, modifier subunit	2.01	3.00 × 10^−4^	0.0973
2	*ABTB2*	ankyrin repeat and BTB (POZ) domain containing 2	2.15	9.88 × 10^−5^	0.0518
3	*LOC344887*	NmrA-like family domain containing 1 pseudogene	2.18	3.00 × 10^−4^	0.088
4	*SRXN1*	sulfiredoxin 1	2.24	1.00 × 10^−4^	0.0525
5	*LIPG*	lipase, endothelial	2.33	1.00 × 10^−4^	0.0549
6	*LUCAT1*	lung cancer associated transcript 1 (non-protein coding)	2.39	5.67 × 10^−5^	0.0408
7	*DLGAP1-AS1*	DLGAP1 antisense RNA 1	2.39	2.00 × 10^−4^	0.0674
8	*SLC7A11*	solute carrier family 7 (anionic amino acid transporter light chain, xc-system), member 11	2.47	9.21 × 10^−5^	0.0493
9	*MIR22HG; MIR22*	MIR22 host gene; microRNA 22	2.58	2.00 × 10^−4^	0.0657
10	*KRTAP2-3*	keratin associated protein 2-3	2.88	3.14 × 10^−5^	0.0296
11	*DHRS9*	dehydrogenase/reductase (SDR family) member 9	2.94	2.06 × 10^−5^	0.0257
12	*ARRDC4*	arrestin domain containing 4	4.15	2.25 × 10^−5^	0.0257

## Data Availability

The dataset has been deposited to NIH Gene Expression Omnibus located at https://www.ncbi.nlm.nih.gov/geo/query/acc.cgi?acc=GSE189183 (approved), accessed on 20 November 2021.

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
