# Peer review of "Triple Isozyme Lactic Acid Dehydrogenase Inhibition in Fully Viable MDA-MB-231 Cells Induces Cytostatic Effects That Are Not Reversed by Exogenous Lactic Acid"

_biomolecules, 2021, doi:10.3390/biom11121751_

Round 1

Reviewer 1 Report

Dr. Elizabeth Mazzio and colleagues reported their recent study for investigating “Lactic Acid Dehydrogenase Inhibition in Triple-Negative Breast Cancer Cells is Associated with Cytostatic Effects and is Independent of the Mitigating Loss of Lactic Acid”. Scientists have experimented with cancer patients infiltrating human plasms with poor prognosis. Although FDA approval for the inhibition of the immune check point for treating TNBC patients, However, improving current low response rates is essential. Most cancers have metabolic plasticity and are divided to highly glycolytic tumors through the Warburg effect. The LDHA and following subsequent lactate accumulation trigger and induce an acidic pericellular microenvironment near tumors. The authors used a triple LDH isoform inhibitor (GNE-140) on MDA-MB-231 cells, according their interesting conclusion suggests that complete inhibition of LDH in cancer cells is not detrimental to cell viability or energy systems. Although the authors offered the point view is not novel (many publications about LDH-A inhibitor were applied in cancer, including TNBC / metastatic TNBC), some analytic results maybe provide the novel potential therapeutic strategies for malignant breast cancer, especially for triple negative breast cancer. Finally, the results of this paper are interesting but some concerns should be addressed to revise:

  • All references have to revise its style, they are difficult to read for readers. Such as Line 30 “…., high recurrence, and low overall survival rates 1,2”. please revise to” ………. overall survival rates [1,2]” or survival rates 1,2
  • Many typing errors, like Line 76 “75 cm2 flasks” and Line 78” 5% CO2” should be revised to” 75 cm2 flasks” and “5% CO2”, the many similar errors let reviewers feel casual attitude in preparing this manuscript.
  • Please explant why GEN-140 has capacity to inhibit LDHA/B/C in the low “nM” range, but all study used almost thousand folds dosage (µM) to treatment into MDA-MB 231 cells?
  • Line 181” The IC50 for lactic acid inhibition was 3.04 uM where the IG50 for cell proliferation was 10.51 uM. Please revise the uM to µM, and IG50? The writers' meaning is IC50? Or IC50
  • Line 89-95, this paragraph has a confusing typography...
  • In Figure 3, The GEN-140 show cytostatic effects of GNE-140 in MDA-MB-231 cells after a 7 day proliferation period, the dosage is too high to inhibit (or kill) MB-231 cells, whether the very high dose to kill the cells directly instead not through LDH inhibition in cancer cells.  

Author Response

Reviewer #2. Dr. Elizabeth Mazzio and colleagues reported their recent study for investigating “Lactic Acid Dehydrogenase Inhibition in Triple-Negative Breast Cancer Cells is Associated with Cytostatic Effects and is Independent of the Mitigating Loss of Lactic Acid”. Scientists have experimented with cancer patients infiltrating human plasms with poor prognosis. Although FDA approval for the inhibition of the immune check point for treating TNBC patients, However, improving current low response rates is essential. Most cancers have metabolic plasticity and are divided to highly glycolytic tumors through the Warburg effect. The LDHA and following subsequent lactate accumulation trigger and induce an acidic pericellular microenvironment near tumors. The authors used a triple LDH isoform inhibitor (GNE-140) on MDA-MB-231 cells, according their interesting conclusion suggests that complete inhibition of LDH in cancer cells is not detrimental to cell viability or energy systems. Although the authors offered the point view is not novel (many publications about LDH-A inhibitor were applied in cancer, including TNBC / metastatic TNBC), some analytic results maybe provide the novel potential therapeutic strategies for malignant breast cancer, especially for triple negative breast cancer. Finally, the results of this paper are interesting but some concerns should be addressed to revise:

Comment 1: All references have to revise its style, they are difficult to read for readers. Such as Line 30 “…., high recurrence, and low overall survival rates 1,2”. please revise to” ………. overall survival rates [1,2]” or survival rates 1,2.

Response 1:  We reformatted the references using end-note.

Comment 2: Many typing errors, like Line 76 “75 cm2 flasks” and Line 78” 5% CO2” should be revised to” 75 cm2 flasks” and “5% CO2”, the many similar errors let reviewers feel casual attitude in preparing this manuscript.

Response 2: Reviewer is correct: This was also noted by reviewer #1. Subscript fonts have been corrected for IC50 CO2, 75 cm2 flasks, cell densities and symbols corrected for μM.

Comment 3: Please explant why GEN-140 has capacity to inhibit LDHA/B/C in the low “nM” range, but all study used almost thousand folds dosage (µM) to treatment into MDA-MB 231 cells?

Response 3: This may be somewhat unclear, but typically when describing enzyme inhibition by IC50 concentrations , this concentration is always  dependent upon enzyme concentrations in Units/ml and supply of substrate and cofactors. Enzyme concentration in recombinant isolated enzyme stuides is not always equal to the LDH concentrations within cells. From our experience , pure enzyme inhibition is often in the low nM ranges , due to the pure enzyme rate, while a cell line with high amount of enzyme such as TNBC has a higher level of LDH with shift the IC50 to the right at high concentration. The inhibition of LDH is important, but what makes this study interesting to us is that, this is the first time in our experience we found a drug that fully blocked production of lactic acid in a cancer cell line.   The concentration of the drug is secondary to analyze exactly what is happening on the molecular level when glycolysis if severely impaired.

Comment 4 : Line 181” The IC50 for lactic acid inhibition was 3.04 uM where the IG50 for cell proliferation was 10.51 uM. Please revise the uM to µM, and IG50? The writers' meaning is IC50? Or IC50 

Response 4: This was also mentioned by Reviewer 1. The subscript corrections have been made throughout the manuscript, and uM to uM. Yes this is correct, we used the IG50 term to identify inhibitory growth concentration at 50% of the untreated controls.

Comment 5: Line 89-95, this paragraph has a confusing typography ..

Response 5: This has been edited to “ After maturation , GNE-140 was added to the wells, and spheroids were grown for another 14 days followed by testing with phase contrast and fluorescence imaging. Imaging was captured with phase contrast and fluorescence imaging using a Biotek Cytation 5 imaging Station and Gen 5.0 software (Bio-Tek, Winooski, VT, USA). “

Comment 6: In Figure 3, The GEN-140 show cytostatic effects of GNE-140 in MDA-MB-231 cells after a 7 day proliferation period, the dosage is too high to inhibit (or kill) MB-231 cells, whether the very high dose to kill the cells directly instead not through LDH inhibition in cancer cells.

Response 6: This is a very good point. These cells are still very much alive but static. We have changed the curve to represent cell count (viable cells) vs control, rather than cell proliferation % control. This drug , very similar to cytostatic drugs like taxol is non -toxic , but cytostatic. Dead cells do not divide, and therefore true cytostatic effects would require live cells , that continue to attempt to divide but can not do so. Like taxol and in Figure 4, cytostatic cells always display an enlarged cytoskeletal structure, from attempts at cell division but failure.  In our past studies, we have like others reported that this coincides with multi-nucleation of large abnormally shaped enlarged cells. This issue is often confused, but cytotoxicity is a very different thing than cytostatic, the latter can only be proven in live cells.  

Submission Date

05 October 2021

Date of this review

24 Oct 2021 01:26:40

Comments and Suggestions for Authors

Reviewer 2 Report

In this study the authors support the conclusion that a multi-LDH isoform inhibitor GNE-140 completely inhibits lactate production without having any effect in cell viability. Adding lactate in the cultures does not compensate for the loss of the LDH enzymatic activity. It is a very  interesting finding, however, there is a concern about the conclusion regarding the viability because the measurements were based on Alamar Blue, a dye which stains only viable cells, and thus apoptotic or pre-apoptotic cells will not be detected. For a more accurate assessment, you could measure viability at different time points such as 24, 48 and 72h by staining with PI and AnnexinV and flow cytometry. Also, it is not clear if an appropriate control treatment with vehicle (DMSO or other) was used. In addition, Figures and figure legends can be significantly improved by providing detailed information and quantitative assessments. Please also consider several suggestions for improving the language.

Major points:

Lines 13-14: “The data show that GNE-140 fully blocks the production of lactic acid, which is directly inverse to a block of glucose utilization”. It is difficult to understand what this phrase suggests, could you make it more clear? Why is this directly “inverse” to a block of glucose utilization? Do you want to say that GNE-140 blocks production of lactate and this also results in a block of glucose utilization and of the glycolytic pathway?

Lines 19-20: “The cytostatic effects appear to occur at the G1/S phase as determined by flow cytometry analysis, and similar transcripts were also reflected by whole transcriptomic data”. I think “transcripts” does not fit well here. Maybe you can say “similar conclusions are also drawn from whole transcriptomic data”.

Line 21: …relatively mild…What do you mean with “mild”?.  In the next sentence, you indicate significant reduction of transcripts related with many critical functions! Is this considered “mild”?

Lines 63-65: You stated that: “In this work, we have examined for the first time the LDH inhibitor drug that can entirely block lactic acid production, GNE-140”. You already referenced the study that discovered and used GNE-140 for the first time (Ref 14, Boudreau et. al. 2016). So this statement doesn’t sound right, could you rephrase? Do you mean first time used with the MDA-MD-231 cells?

Line 97: In the materials’ section 2.3 for Imaging, a methodology for Live cell imaging is described. Live cell imaging does not involve fixation, but further down, in Line 104, the cells are fixed. Also, Propidium iodide (PI) is a viability dye (permeable only to dead cells) staining nucleic acids, how is it used for cytoskeletal morphology? Can you clarify?

Line 174: Here, the conclusion is made that there was no effect on cell viability. However, as is described in the materials, Alamar Blue stains viable cells. So, how about the cells that are apoptotic or pre-apoptotic? You could assess viability at different time points such as 24, 48 and 72 h by staining with PI and AnnexinV (without prior fixation) and flow cytometry, or Casp3 as in ref 14 Boudreau et al NatChemBiol 2016. This approach will help confirm your findings and give a better idea about the apoptotic effects caused by GNE-140 (please also include a vehicle control, DMSO or ethanol depending on which one you used to dissolve the compound).  

Lines 175-176: Data were also acquired on oxygen consumption and mitochondrial potential (Janus green and MitoTracker® Red, respectively), reflecting no change between controls and GNE-140 treatment (data not shown). You should show the data and also include in the materials how this dyes are used and measured. Is it by flow cytometry? You can show the flow graphs.

Line 179 and Figure 3: Which other dyes are shown in the images besides DAPI? Phalloidin-iFluor488? Please provide a detailed legend. In the higher drug concentrations, there are fewer cells present, could that mean that the GNE-140 caused significant apoptosis and only few cells have survived? Is this number of cells remaining equal to the number of cells initially plated or is it lower? How is the enlargement of actin cytoskeleton indicated, it is not clear. You could also add a quantification assessment.

Figure 4: Is there any quantification assessment?

Figure 5. You indicate (+) Lactate used in the legend and then Media DL-lactate in the figure. Which one of the two was used? In the text it is described that Alamar Blue viability dye is used for the right panel which in the Figure indicates proliferation instead of viability. How do you conclude that there is no cytotoxic effect? Is it possible that as the cells die they may be becoming more permeable to the dye, and falsely indicate viable cells? As in the previous comment, to accurately assess viability, you should perform flow cytometry using PI together with Annexin V (indicates pro-apoptotic cells) at various time points under non-fixed conditions, with immediate live cell data acquisition.  Again, as previously mentioned, it is not clear whether you use vehicle (e.g. DMSO) as control for GNE-140. The same applies for the other figures as well.

Line 298: In the study from Boudreau et al (Fig 1F), it was shown that GNE-140 causes a significant increase of Casp3+ cells and % sub-2N cells after days ≈2-3, how does this align with your findings with TNBC cells showing no effect in viability?

Line 316: It should also be discussed that the major pathway of effector T cell function is glycolysis, and using a global LDH inhibitor could potentially severely impair anti-tumor T cell responses.

Figure 10. This figure includes several undefined parameters (e.g. what is model 1D0n DSD, Total B.A.D., RCS, Dip?, %CV).

Suggestions/corrections for improving the text:

Line 43: …is integral in its acidity related role of to drive metastasis and immune evasion. You may modify to: …is integral in its acidification-related role of driving metastasis and immune evasion.

Line 45: …it became more evident… Here you may say “it was very puzzling that there were no changes…”

Lines 46-50: You may modify to: Our attempts to block lactic acid production by 1) LDHA gene silencing and 2) LDHA inhibitors derived from high-throughput screening, failed 10, 11.

Line 50: The ability to block lactic acid was not forthcoming in our experiments leading to the supposition that…You may replace this phrase with: These findings led to the supposition that….

Line 56: May be better to list the refs chronologically, Zdralevic, M. et al. (2018) and then Mazzio et al. (2020)

Line 61:Use “GNE-140” for accuracy.

Line 65: You may change: …maintained experiments… to …performed experiments…

Lines 74-89: Line spacing is different.

Line 90 and below: Merge the paragraph formatting with rest of the text.  

Line 81 and 82: Correct 105 to 105.

Line 93: You may modify as: …for another 14 days, followed by phase contrast and fluorescence imaging…

Line 121: Correct power superscripts: 75 cm2 flasks at 0.5 ×106 cells/ml

Line 122: The units do not show up, is it 50 µM GNE-140?

Line 134: Is the number 2.1 correct?

Line 170: Correct to: …data were collected…

Line 171: Keep consistent formatting of figure indicators in whole manuscript (Figure 2A or Fig.2A, bold or not etc…)

Line 172: …GNE-140 can effectively block lactic acid production, inverse to hampered utilization of glucose… Does this mean the GNE-140 blocked both lactic acid production and glycolysis? The phrase “inverse to hampered utilization of glucose” is confusing.

Figures and legends: You may change uM to µM.

Line 190: Correct to …(10 µM)…

Line 202: …in the presence “of” 120 µM GNE-140…

Line 225: Provide definition for DEGS.

Line 264-267: “The data in this study investigates the consequences of inhibiting multiple isoforms of LDH, with total suppression of lactic acid production, severely stunted glucose use, and an impediment to what is termed the anaerobic glycolytic pathway”. You can modify as:

Our study investigated the consequences of inhibiting multiple isoforms of LDH with a single reagent (GNE-140), on MDA-MB-231 TNBC cells, showing total suppression of lactic acid production, severely stunted glucose utilization, and an impediment to what is termed the anaerobic glycolytic pathway.

Line 269: You may remove: “within our research group”.

Not sure if it is random, but some space gaps appear in the text:

Lines: 13, 23, 30, 33, 44, 48, 50, 59, 92, 98, and more…

Author Response

Reviewer: “In this study the authors support the conclusion that a multi-LDH isoform inhibitor GNE-140 completely inhibits lactate production without having any effect in cell viability. Adding lactate in the cultures does not compensate for the loss of the LDH enzymatic activity. It is a very  interesting finding, however, there is a concern about the conclusion regarding the viability because the measurements were based on Alamar Blue, a dye which stains only viable cells, and thus apoptotic or pre-apoptotic cells will not be detected. For a more accurate assessment, you could measure viability at different time points such as 24, 48 and 72h by staining with PI and AnnexinV and flow cytometry. Also, it is not clear if an appropriate control treatment with vehicle (DMSO or other) was used. In addition, Figures and figure legends can be significantly improved by providing detailed information and quantitative assessments. Please also consider several suggestions for improving the language.

Major points:

Comment 1 :Lines 13-14: “The data show that GNE-140 fully blocks the production of lactic acid, which is directly inverse to a block of glucose utilization”. It is difficult to understand what this phrase suggests, could you make it more clear? Why is this directly “inverse” to a block of glucose utilization? Do you want to say that GNE-140 blocks production of lactate and this also results in a block of glucose utilization and of the glycolytic pathway? 

Response 1: Reviewer is correct: This is exactly what we are trying to say, but we were having trouble wording this. Thank you and Line 13 has been edited according to this suggestion.The sentence now reads “The data shows that GNE-140 fully blocks the production of lactic acid, which also results in a block of glucose utilization and severe impedance of the glycolytic pathway.”

Comment 2 : Lines 19-20: “The cytostatic effects appear to occur at the G1/S phase as determined by flow cytometry analysis, and similar transcripts were also reflected by whole transcriptomic data”. I think “transcripts” does not fit well here. Maybe you can say “similar conclusions are also drawn from whole transcriptomic data”.

Response 2. We have integrated the reviewers suggestion as it accurately describes what we are trying to convey. This sentence now reads: “The cytostatic effects occur at the G1/S phase as determined by flow cytometry analysis, and similar conclusions are also drawn from whole transcriptomic data.”

Comment 3: Line 21: …relatively mild…What do you mean with “mild”?.  In the next sentence, you indicate significant reduction of transcripts related with many critical functions! Is this considered “mild”?

Response 3:  The term mild was used, because while the effect of the drug on glycolysis was remarkable , we expected to see severe transcriptome differential changes by way of transcript count, which was not the case. This is unusually low compared to most anti-cancer drugs, we have run  in this facility. We have changed the text to read as follows:  “The effects of GNE-140 on the whole transcriptome was relatively mild (22 up-regulated differential expressed transcripts (DEGs) / 77 down regulated DEGs out of 48,226 evaluated. Down-regulated DEGS centered around a loss of genes related to mitosis, cell cycle, GO/G1; G1/S transition, and DNA replication.”

Comment 4. Lines 63-65: You stated that: “In this work, we have examined for the first time the LDH inhibitor drug that can entirely block lactic acid production, GNE-140”. You already referenced the study that discovered and used GNE-140 for the first time (Ref 14, Boudreau et. al. 2016). So this statement doesn’t sound right, could you rephrase? Do you mean first time used with the MDA-MD-231 cells?

Response 4. Reviewer is correct. We have edited the text to read “In this work, we  investigate the effects of GNE-140 , an efficiently multi-isoform LDH inhibitor, on biochemical parameters and whole transcriptomic response in a TNBC cell model.  “

Comment 5: Line 97: In the materials’ section 2.3 for Imaging, a methodology for Live cell imaging is described. Live cell imaging does not involve fixation, but further down, in Line 104, the cells are fixed. Also, Propidium iodide (PI) is a viability dye (permeable only to dead cells) staining nucleic acids, how is it used for cytoskeletal morphology? Can you clarify?

Response 5: Live cell imaging was used for staining of spheroids, and also the monolayer for cell count in extended proliferation studies. Morphological cytoskeletal staining was acquired using phalloidin in fixed permeabilized monolayers.  This section had been edited to clarify imaging of 2D vs 3D

Comment 6. Line 174: Here, the conclusion is made that there was no effect on cell viability. However, as is described in the materials, Alamar Blue stains viable cells. So, how about the cells that are apoptotic or pre-apoptotic? You could assess viability at different time points such as 24, 48 and 72 h by staining with PI and AnnexinV (without prior fixation) and flow cytometry, or Casp3 as in ref 14 Boudreau et al NatChemBiol 2016. This approach will help confirm your findings and give a better idea about the apoptotic effects caused by GNE-140 (please also include a vehicle control, DMSO or ethanol depending on which one you used to dissolve the compound).  

Response 6. Agree - common misconception. We think the best way to clarify this, is to change Figure 2B to Cell Count as % Untreated controls rather than Cell Proliferation as % untreated controls. Working extensively with a large number of LDH inhibitors in this study and in the past, we could not document a single instance where inhibition of this enzyme caused cell death (either apoptosis or necrosis). This was finally clarified using double knockdowns in a prior study, where absence of production of lactic acid and LDH expression - was completely non toxic (cell division was impacted). In fact blocking LDH make cells stronger and more resilient to a wide variety of anti-cancer drugs. This is because typically pyruvate is simply rerouted to the mitochondrial  and there is a surge of oxidative ATP production. Evidence of a lack of toxicity is also found in this work, by the 7 day proliferation study. These cells are fully alive- even though they are not dividing.  There is a common misconception regarding the very distinct differences between cytostatic and cytotoxic compounds, and the experiments which make this clear. Toxic compounds predominate where dead cells do not divide, and the IG50 and LC50 curves are identical or in very close proximity. True and real anti-mitotic effects can only be discerned in fully viable cells, where there is no cell death , as in the case for GNE-140 and taxol, which becomes evident both in long term  extended proliferation studies as well as short term 24 hour toxicity studies in a high density of cells. If you look at the differential shown in Figure 2 , this fact becomes very clear. There is no toxicity in a high density of cells over 100uM, where proliferation is completely compromised at low uM ranges in fully viable cells.  

Comment 7: Lines 175-176: Data were also acquired on oxygen consumption and mitochondrial potential (Janus green and MitoTracker® Red, respectively), reflecting no change between controls and GNE-140 treatment (data not shown). You should show the data and also include in the materials how this dyes are used and measured. Is it by flow cytometry? You can show the flow graphs.

Response 7:   We added mitochondrial data to Figure 2A, overlapping data on cell viability , lactic acid  and updated the methods section for imaging to include mitotracker red assay.  

Comment 8: Line 179 and Figure 3: Which other dyes are shown in the images besides DAPI? Phalloidin-iFluor488? Please provide a detailed legend. In the higher drug concentrations, there are fewer cells present, could that mean that the GNE-140 caused significant apoptosis and only few cells have survived? Is this number of cells remaining equal to the number of cells initially plated or is it lower? How is the enlargement of actin cytoskeleton indicated, it is not clear. You could also add a quantification assessment.

Response 8: This is very observant of Reviewer 1. DAPI is the only dye, but the yellow perimeter is the analytical masking by the cytation 5 software to count cells. This needs to be clarified.   Figure 3 now reads “Figure 3. Cytostatic effects of GNE-140 in MDA-MB-231 cells after a 7 day proliferation period.  Images represent nuclear cell count in viable cells using DAPI staining, with cell count analytical masking (yellow outline) used by Cytation 5 Analytical Imaging Software to count cells.“

Comment 9: Figure 4: Is there any quantification assessment?

Response 9: Figure 4 is just  to show morpholgical change in cytostatic cells. Figure 4 is a complement to Figure 3, and Figure 3 corroborates the cytostatic changes by Alamar Blue as collaborated in Figure 2B.

Comment 10: Figure 5. You indicate (+) Lactate used in the legend and then Media DL-lactate in the figure. Which one of the two was used? In the text it is described that Alamar Blue viability dye is used for the right panel which in the Figure indicates proliferation instead of viability. How do you conclude that there is no cytotoxic effect? Is it possible that as the cells die they may be becoming more permeable to the dye, and falsely indicate viable cells? As in the previous comment, to accurately assess viability, you should perform flow cytometry using PI together with Annexin V (indicates pro-apoptotic cells) at various time points under non-fixed conditions, with immediate live cell data acquisition.  Again, as previously mentioned, it is not clear whether you use vehicle (e.g. DMSO) as control for GNE-140. The same applies for the other figures as well.

Response 10:

  • Used (+) lactic acid (legend is now corrected)…and
  • Note: Left hand margin is cell count using Alamar blue , and the images on the right is DAPI nuclear cell count. There was no effect of lactic acid on cell proliferation, which is an important aspect of this study. We felt we had to show this at the effective concentration of 10uM (Cytostatic) with no toxic effects (Figure 2A) +- lactic acid. Note: Cytotoxicity studies , we always conduct at high cell density at 24 hours. Cytostatic studies, we always conduct at around 6-7 day endpoint, where all cells are viable, but the cells are not dividing. We commented on this also in Comment 6. There is a common misconception about this as there is a very big difference between a cytostatic anti-cancer drug and one that is cytotoxic. Taxol is the best example, where this drug is not cytotoxic at all (from our experiments), which is why it is very easy to elucidate non dividing live cells at 7 days. The morphological structure of anti-mitototics as we also show in this study is extended large abnormal cytoskeletal structures , as cells continue to attempt to divide ,and fail. If you look close at Figure 4, we can see this as cells are unable to divide.

Comment 11: Line 298: In the study from Boudreau et al (Fig 1F), it was shown that GNE-140 causes a significant increase of Casp3+ cells and % sub-2N cells after days ≈2-3, how does this align with your findings with TNBC cells showing no effect in viability?

Response 11: Agree - common misconception.This depends on the definition of cell death by the researcher and the concentrations used. Typically , in our facility - we define cell death by energy, which occurs as a result of apoptosis, and / or necrosis always circumscribing losses in membrane potential, ATP, mitochondrial function , cell viability, lactic acid and transcript changes which reflect apoptosis. As per this definition, we are not showing any significant adverse effects by GNE-140, with the primary outcome on cell proliferation. Why they are showing caspase 3+, we are not sure - we are not seeing cell death or caspase 3+ expression in the microarray data.  

Comment 12: Line 316: It should also be discussed that the major pathway of effector T cell function is glycolysis, and using a global LDH inhibitor could potentially severely impair anti-tumor T cell responses. 

Response 12: This is a good point and it could  be a potential counter indication for the use of LDH inhibitors. We will add some information on this.

Comment 11: Figure 10. This figure includes several undefined parameters (e.g. what is model 1D0n DSD, Total B.A.D., RCS, Dip?, %CV).

Response 11: The legend now includes the abbreviations.

Suggestions/corrections for improving the text:

Line 43: …is integral in its acidity related role of to drive metastasis and immune evasion. You may modify to: …is integral in its acidification-related role of driving metastasis and immune evasion.

Response: has been edited to read as suggested.

Line 45: …it became more evident… Here you may say “it was very puzzling that there were no changes…”

Response: has been edited to read as suggested.

Lines 46-50: You may modify to: Our attempts to block lactic acid production by 1) LDHA gene silencing and 2) LDHA inhibitors derived from high-throughput screening, failed 10, 11.

Response: has been edited to read as suggested.

Line 50: The ability to block lactic acid was not forthcoming in our experiments leading to the supposition that…You may replace this phrase with: These findings led to the supposition that….

Response: has been edited to read as suggested.

Line 56: May be better to list the refs chronologically, Zdralevic, M. et al. (2018) and then Mazzio et al. (2020) 

Response: References have been switched

Line 61:Use “GNE-140” for accuracy.

Response:  Checked to make sure all instances are referenced by GNE-140.

Line 65: You may change: …maintained experiments… to …performed experiments…

Response: This sentence no longer exists after editing the last sentence of introduction.

Lines 74-89: Line spacing is different.

Response: I believe we fixed this, but double check with the editor on this.

Line 90 and below: Merge the paragraph formatting with rest of the text.  

Response: That is exactly what we did to fix the line spacing problem.

Line 81 and 82: Correct 105 to 105.

Response: This as been corrected, to include subscript as suggested by reviewer.

Line 93: You may modify as: …for another 14 days, followed by phase contrast and fluorescence imaging…

Response: has been edited to read as suggested.

Line 121: Correct power superscripts: 75 cm2 flasks at 0.5 ×106 cells/ml

Response: has been edited to read as suggested throughout.

Line 122: The units do not show up, is it 50 µM GNE-140?

Response: Yes. µM has been added.

Line 134: Is the number 2.1 correct?

Response: Yes this refers to the 2.1 in the affymetrix 2.1 array. I think for clarity, it is best to remove - so it has been removed from the subtitle.

Line 170: Correct to: …data were collected…

Response: has been edited to read as suggested.

Line 171: Keep consistent formatting of figure indicators in whole manuscript (Figure 2A or Fig.2A, bold or not etc…)

Response: has been edited to read as suggested.

Line 172: …GNE-140 can effectively block lactic acid production, inverse to hampered utilization of glucose… Does this mean the GNE-140 blocked both lactic acid production and glycolysis? The phrase “inverse to hampered utilization of glucose” is confusing.

Response: Yes, this was hard for us to explain. We know that lactic acid is no longer produced, and glucose is no longer used.. but glycolysis has a large number of biological reactions, so we were a bit hesitant to suggest this. But yes , the over rate of the glycolytic pathway appears slowered.

Now reads” The data demonstrate that GNE-140 can effectively block lactic acid production, inverse to hampered utilization of glucose, suggesting an attenuated rate of glycolysis. This manipulation had no adverse effects on cell viability, but there was a loss of cell division as observed by a 7-day end-point proliferation study.”

Figures and legends: You may change uM to µM.

Response: has been edited to read as suggested.

Line 190: Correct to …(10 µM)…

Response: has been edited to read as suggested.

Line 202: …in the presence “of” 120 µM GNE-140…

Response: has been edited to read as suggested.

Line 225: Provide definition for DEGS.

Response: We introduce the term differentially expressed genes and abbreviation in the abstract (DEG) , also by answering question 3 (above) .

Line 264-267: “The data in this study investigates the consequences of inhibiting multiple isoforms of LDH, with total suppression of lactic acid production, severely stunted glucose use, and an impediment to what is termed the anaerobic glycolytic pathway”. You can modify as:

Our study investigated the consequences of inhibiting multiple isoforms of LDH with a single reagent (GNE-140), on MDA-MB-231 TNBC cells, showing total suppression of lactic acid production, severely stunted glucose utilization, and an impediment to what is termed the anaerobic glycolytic pathway.

Response: The sentence has been replaced with the reviewers revision.

Line 269: You may remove: “within our research group”.

Response: has been edited to read as suggested.

Not sure if it is random, but some space gaps appear in the text:

Lines: 13, 23, 30, 33, 44, 48, 50, 59, 92, 98, and more…

Response: This has been noted, and we may request assistance from the editor.

Comments and Suggestions for Authors

Reviewer 3 Report

The manuscript “Lactic Acid Dehydrogenase Inhibition in Triple-Negative Breast Cancer Cells is Associated with Cytostatic Effects and is Independent of the Mitigating Loss of Lactic Acid” describes using triple LDH isoform inhibitor 12 (A, B &C) (GNE-140) on MDA-MB-231 triple-negative breast cancer cells. GNE-140 blocks the production of lactic acid, which is directly inverse to a block of glucose utilization in MDA-MB-231 cells. Moreover, authors analyze the effects of GNE-140 on the whole transcriptome and show that there is some element linking LDH enzyme activity with the cell cycle.

Comments:

  1. Authors study the effects of GNE-140 in only one cell line, so it is difficult to make a conclusion about all triple-negative breast cancer cells. Authors need to show these effects on more than one cell line.
  2. The effects of GNE-140 on cell cycle progression in MDA-MB-231 cells by Flow Cytometry need to be improved. Experiments need to be repeated with different concentrations of GNE-140 and need to show statistics.
  3. Authors need to confirm the date of the whole transcriptome for several up-regulated and down-regulated transcripts by qPCR

Author Response

Reviewer #3:The manuscript “Lactic Acid Dehydrogenase Inhibition in Triple-Negative Breast Cancer Cells is Associated with Cytostatic Effects and is Independent of the Mitigating Loss of Lactic Acid” describes using triple LDH isoform inhibitor 12 (A, B &C) (GNE-140) on MDA-MB-231 triple-negative breast cancer cells. GNE-140 blocks the production of lactic acid, which is directly inverse to a block of glucose utilization in MDA-MB-231 cells. Moreover, authors analyze the effects of GNE-140 on the whole transcriptome and show that there is some element linking LDH enzyme activity with the cell cycle.

Comment 1: Authors study the effects of GNE-140 in only one cell line, so it is difficult to make a conclusion about all triple-negative breast cancer cells. Authors need to show these effects on more than one cell line. 

Response 1: This is true. Originally Boudreau et al., studied GNE-140 in various cancer cell lines , showing this drug is effective to block human LDHA ,B and C resulting in a full impedance of lactic acid and glycolysis (supplemental data in that studies, shows glucose is not being consumed rapidly). Rapid glycolysis is a basic tenement to almost all human cancers. In this work, we extend on Boudreus work in a second cell line (breast: MDA-MB-231 cells) by GNE-140, however our findings in this (the current work) were previously confirmed in human colon cancer cells (colon: LS174T) by double knock out LDHA and B, all showing the same identical results: 1) cells survive, 2) produce ample ATP 3) can no longer rapidly consume glucose 4) a consistent observation of slower cell division . In this study , we elaborate and confirm more on aspects of this, but this work will likely be applicable to all types of human cancer. We are requesting that this body of work , be considered as presented. Thank you kindly for your consideration.      

Comment 2: The effects of GNE-140 on cell cycle progression in MDA-MB-231 cells by Flow Cytometry need to be improved. Experiments need to be repeated with different concentrations of GNE-140 and need to show statistics. 

Response 2: AGREE: In the past 3 weeks, we worked with Brad Larsen from  Biotek, Agilent - to conduct digital cytometry experiments in 96 well plates using our newly acquired Cytation 5 imaging system. The data is similar to the flow, and now added to the manuscript, which  we included at several concentrations, n=3 with statistics.   We also added Andrew Barnes as a co-author who worked with Dr. Mazzio and Brad Larsen to establish protocols required to carry out this work.

Comment 3: Authors need to confirm the data of the whole transcriptome for several up-regulated and down-regulated transcripts by qPCR.

Response 3: This is true, however the transcriptome showed very minor changes, 77 down 12 up / of over 48000 transcript tested: where down-regulated genes centered on loss of cell cycle at G0/G1 phase, which we corroborate by a cell proliferation study and dual cytometric analysis. While we agree, the affymetrix is a highly precise instrument, and the totality of the data is in agreement where the most important finding of this work, has very little to do with gene expression.

The major finding of this work is that lactic acid (exogenous) can not reverse the anti-mitotic effects of an LDHA,B and C inhibitor. This means that the rate  of glucose - lactic acid conversion by LDH multiple forms controls cell cycle (independent of lactic acid).We are requesting that this body of work , be considered as presented. Thank you kindly for your consideration.      

Submission Date

05 October 2021

Date of this review

20 Oct 2021 22:07:22

Round 2

Reviewer 2 Report

I would like to thank the authors for addressing all my previous comments. The manuscript has been significantly improved.

Reviewer 3 Report

Since the authors studied the effect of the inhibitor on only one cell line (MDA-MB-231), it is worth to refer MDA-MB-231 in the title of the article. 

Agree: Title has been changed to include this one particular cell lines.